# Evaluation of Nutritive Values through Comparison of Forage Yield and Silage Quality of Mono-Cropped and Intercropped Maize-Soybean Harvested at Two Maturity Stages

**Maw Ni Soe Htet [1], Jiang-Bo Hai [1], Poe Thinzar Bo [1], Xiang-Wei Gong [1], Chun-Juan Liu [1], Ke Dang [1], Li-Xin Tian [1], Rab Nawaz Soomro [2], Khaing Lin Aung [3] and Bai-Li Feng [1],***

[1] State Key Laboratory of Crop Physiology and Tillage in Northwestern Loess Plateau, College of Agronomy, Northwest A&F University, Yangling District, Xianyang 712100, China; mawni2018071063@nwafu.edu.cn (M.N.S.H.); haijiangbo@nwafu.edu.cn (J.-B.H.); poethinzarbo@nwafu.edu.cn (P.T.B.); gongxiangwei@nwsuaf.edu.cn (X.-W.G.); liuchunjuan2016@nwafu.edu.cn (C.-J.L.); dangk18@nwafu.edu.cn (K.D.); 2019060003@nwafu.edu.cn (L.-X.T.)

[2] Key Laboratory of Animal Nutrition and Healthy Feeding, College of Animal Science and Technology, Northwest A&F University, Yangling District, Xianyang 712100, China; nawaz@nwsuaf.edu.cn

[3] State Key Laboratory of Vegetable Crops Stress Biology for Arid Areas, College of Horticulture, Northwest A&F University, Yangling District, Xianyang 712100, China; kla2019071014@nwafu.edu.cn

\* Correspondence: fengbaili@nwsuaf.edu.cn; Tel.: +86-029-8708-2889

**Abstract:** Maize and soybean intercropping is a cereal-legume intercropping pattern that not only increases grain yield but also improves the nutritional value of silage. Experiments were conducted in the summer season to compare the yield and nutritional composition of the forage and silage quality of mono-cropped maize and intercropped maize-soybean harvested at two stages of maturity. The main treatments were one sole crop maize (SM) and four maize-soybean intercropping patterns (one-row maize to one-row soybean (1M1S), one-row maize to two-row soybean (1M2S), one-row maize to three-row soybean (1M3S), and two-row maize to one-row soybean (2M1S). The crops were harvested when the maize reached the milk (R3) and maturity (R6) stages. Results indicated a significant increase in the fresh biomass and dry-matter production of maize fodder alone compared with those of maize intercropped with soybean fodder. After 60 days of ensiling period, silage samples were analyzed for pH, organic acids, dry matter, crude protein (CP), ether extract, neutral detergent fiber (NDF), acid detergent fiber (ADF), and other mineral compositions. All intercropped silages showed higher CP values (1M1S, 12.1%; 1M2S, 12.2%; 1M3S, 12.4%; and 2M1S, 12.1%) than the SM silage (8.7%). Higher organic acids were produced in 1M3S than in the other silages. Correlation data showed that CP was highly correlated with lactic acid but negatively associated with crude fiber, nitrogen-free extract, and NDF. Thus, the intercropping of maize and soybean silage is recommended due its enhanced crop production, nutritional values in dairy animals, and prolific animal feedings and because it was scientifically evaluated as a feed stuff. This study indicated that 1M3S was the most preferable among intercropped silages in terms of nutrient composition.

**Keywords:** maize-soybean; maturity; fermentation; nutrient composition; silage quality

---

## 1. Introduction

Many crops are currently being cultivated in China, but most areas and regions are divided, especially for maize production and cultivation, due to the nutritional and market values of these crops. Maize (*Zea mays* L.), a major cereal grain, is widely cultivated in China's food system. In the past few years, considerable attention has been paid to its feeding value due to a shortage in roughage. Maize has the potential to supply large amounts of energy-rich forage for animal diets, and its fodder could safely be fed at all growth stages without any danger of oxalic acid, or prussic acid toxicity as in the case of sorghum [1]. Developed countries recently recommended maize silage for dairy animals,

fattening animals, and young growing animals for enhanced physical production due to its nutritional aspects. Therefore, Bilal et al. [2] reported that maize silage became a major important forage source for dairy cows throughout the world. Ivan et al. [3] and Uher et al. [4] reported that maize silage is poor in protein content (8.8%) compared with legume silage. However, legume materials are not easy to ensile because of their high buffering capacity (BC) and low level of water-soluble carbohydrate (WSC) [5]. Therefore, protein-rich legume and high-energy maize silage could be ensiled to upgrade nutrient composition [6]. Dairy farming, small ruminant farming, and commercial animal farming prefer maize silage as a rich nutritional source for enhanced production.

Intercropping has numerous advantages, such as increased total grain yield, enhanced soil conservation, increased land use efficiency, and further utilization of resources [7,8]. It shows greater forage production performance than mono-cropping, and it is a feasible option for forage production [9]. Some studies have reported that intercropping of maize with legumes for silage is a suitable strategy to improve crude protein (CP) and the other nutritional qualities of silages [10,11]. The new technique of intercropping is famous and preferred for modern agricultural farming due to the benefits of its producing two crops in the same place, and under similar same resources. In intercropped silage, the optimum harvesting time is required to obtain increased nutritional composition. Maize provides a high yield in terms of dry matter (DM), but it produces forage with low protein content. Some studies have also investigated that protein is needed, not only for the growth and milk production of livestock, but also for the digestion of feed by rumen bacteria in ruminant animals [12,13]. Javanmard et al. [11] pointed out that the intercropped maize with different legumes forage had increased the DM yield (DMY) and CP yield (CPY) compared with that of sole maize (SM). Soybean and maize are the choice for intercropping to enhance the production yield of crops. Altinok et al. [14] reported that the use of maize grown for ensilaging and the seeding of soybean with maize in alternate rows as one maize + one soybean or one maize + two soybean highly increased the silage quality and CP content. Some studies have concluded that the intercropping of maize and cowpea resulted in more CP content than maize sole cropping [3]. Intercrops of maize and legumes for silage showed higher CP content, fiber and lactic acid (LA) concentrations than the ensilage of mono-cropped SM [10,15].

Due to optimum nutritive composition and quality, predicting the best harvest time of intercropping is a critically important issue in intercropped silage [16,17]. Marius and Suyash [18] reported that harvesting time is essential in order to achieve not only the best nutritional value, but also to reduce mold contamination risk on the field. However, only limited information on the nutrient composition of forage and silage of maize-soybean intercrop silage is available. Therefore, the present study aims to evaluate the yield and nutrient composition of the forage and silage quality of SM and maize intercropped with soybean as silage for dairy animals.

## 2. Materials and Methods

### 2.1. Experimental Site

This experiment was conducted during the summer season at the North campus experimental area (108°5′ E, 34°18′ N) of Northwest Agriculture and Forestry University, Yangling, Shaanxi Province, China. The climate of Yangling is characterized by a semi-humid climate type with distinctive seasons; the average annual temperature is 12.9 °C, and annual rainfall is 660 mm. The meteorological data of the experimental site are given in Figure 1. The soil is classified as Loess soil [19]. The land preparation methods used in this experiment were ploughing and harrowing. Before sowing, the soil samples were taken from ten randomly selected points within the experimental sites, at a depth of 20–40 cm using a soil auger. The samples were air-dried, ground, sieved, and analyzed for their chemical properties [20]. The values of observed soil chemicals were 0.63 g kg$^{-1}$ of total nitrogen, 5.5 mg kg$^{-1}$ of available phosphorus, 13.3 g kg$^{-1}$ of organic matter, 108.3 mg kg$^{-1}$

of available potassium, and a pH of 8.3. The previous crop was winter wheat, which was harvested on May 20th of each year. Afterwards, wheat straw was removed from the field.

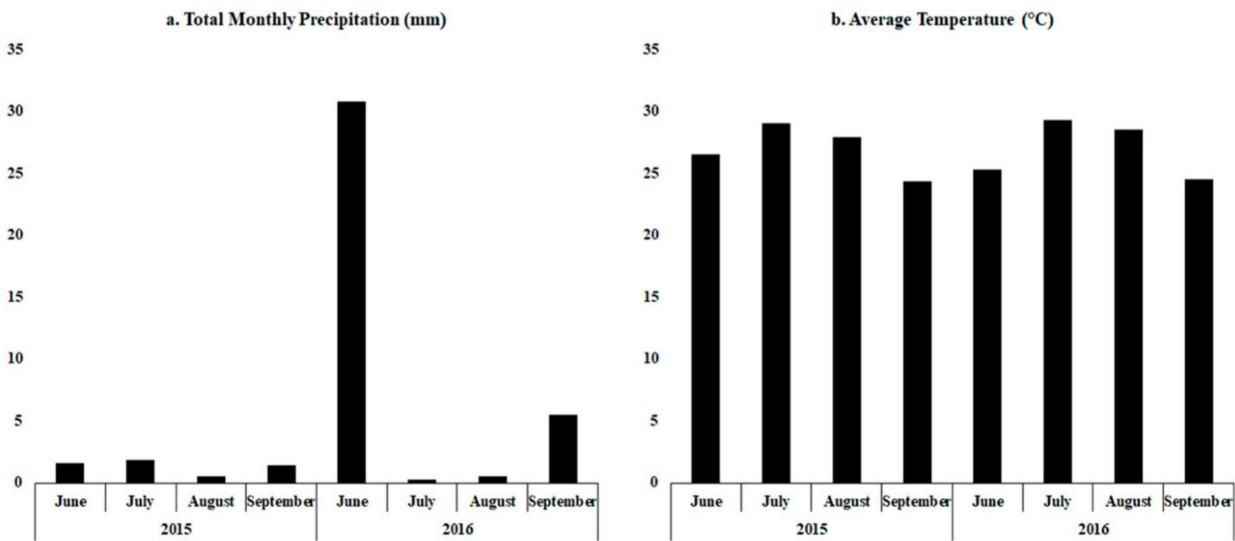

**Figure 1.** Meteorological data during both cropping seasons: (**a**) monthly precipitation, and (**b**) monthly average temperature.

### 2.2. Experimental Treatments and Design

The summer season was ideal for this intercropping because of the seasonal rotational number of crops and suitable sowing time for maize. The summer maize (*Zea mays* L.) local variety recognized as "Zheng Dan 958" was used in this study. It is red in color, and takes 90–110 days to reach the maturity stage. Its seed was supplied by the Seed Company Agricultural Technology Extension Station. The soybean (*Glycine max* L. Zao Huang) crop variety named as a local and annual variety, which matures at 60–75 days, was also used. These soybeans were obtained from a farmer at the Seed Company Township Station. The experiment was a split plot design in a randomized complete block design replicated four times. The main plot treatments were mono-crop and intercropped forages and silages, whereas, maturity was assigned to the sub plots. The individual plot size was 12 m × 5 m. The experimental planting patterns of maize and intercropped maize-soybean are schematized in Figure 2. The experimental treatments consisted of maize seeded as a sole crop (SM) and intercropped with soybean with different planting structures as follows: one-row maize to one-row soybean (1M1S), one-row maize to two-row soybean (1M2S), one-row maize to three-row soybean (1M3S), and two-row maize to one-row soybean (2M1S).

### 2.3. Plant Sampling and Fodder Production

First, maize was seeded on 12 June 2015, and 13 June 2016. After 2 weeks of maize sowing, soybean was seeded on 27 June 2016 and 28 June 2017. The maize and soybean were spaced at 70 cm × 25 cm and 30 cm × 15 cm, with a population of approximately 114,285 and 666,667 plants per hectare, respectively. The experimental site was ploughed to 0.2–0.3 m in depth after the removal of winter wheat straw, followed by harrowing prior to drilling the trial. All plots were fertilized with the same amount of fertilizer before sowing, containing 70 kg of N ha$^{-1}$, 70 kg of P$_2$O$_5$ ha$^{-1}$, and 70 kg of K$_2$O ha$^{-1}$. The seeds of maize and soybean were sown to a depth of approximately 7 and 5 cm, respectively, by hand on 26 June 2016. The seed rates of 10 and 38 seeds of maize and soybean, respectively, per m$^2$ were sown to allow for thinning down to an approximate plant population of 6.7 and 20 plants per m$^2$. Soybean seeds were inoculated with appropriate commercial rhizobium before planting. Neither herbicides nor insecticides were used. Hand weeding by hoe was performed once the maize was approximately 30 cm in height. During the experimental period, the field was irrigated three times with a 30-day interval. The fresh

fodders were manually harvested in both growing seasons, when the maize reached the milk (R3) and maturity (R6) stage of maturity according to Htet et al. [21]. At R3 and R6 stages, 1 m² in three sampling areas was harvested for the determination of fresh biomass yield and further evaluations of nutritional composition.

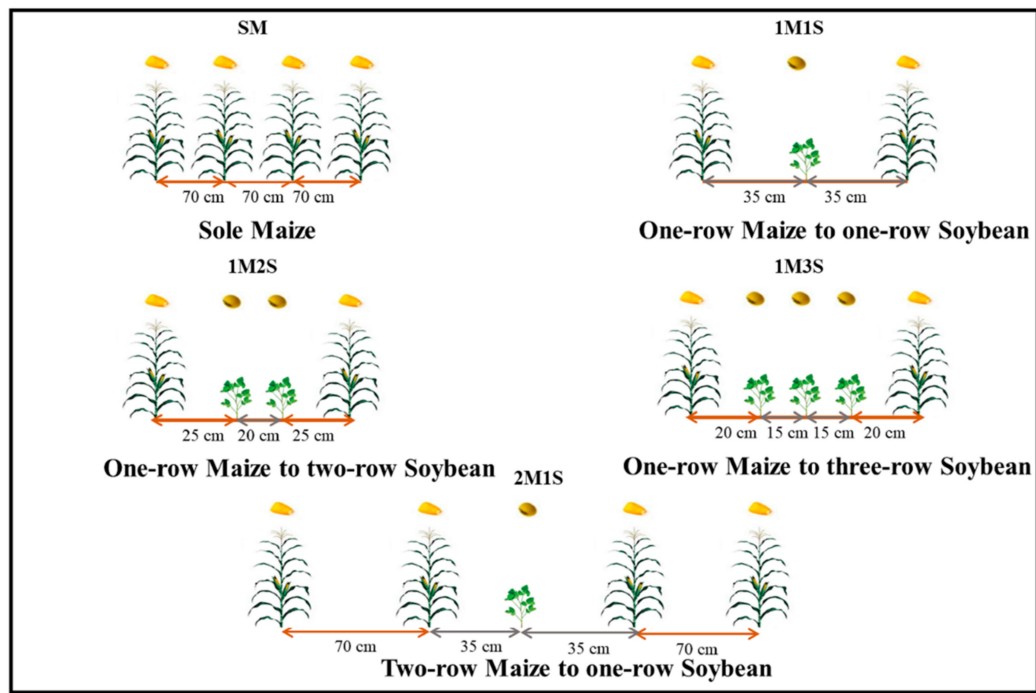

**Figure 2.** Schematic illustration of row layout of maize and intercropped maize-soybean in the experimental planting patterns.

The harvested maize and soybean green fodder were chopped into approximately 3–4 cm in length with a power chaff cutter (JB 400, Gujarat, India) and weighed separately using a spring balance. Then, forage yield was converted into t ha$^{-1}$. A random sample of 500 g fresh forage was collected from each plot for each species, weighed, and dried for 48 h in an oven (GZX-9140MBE, Shanghai Boxun Co., Ltd., Shanghai, China) at 80 °C for determination of DMY. The DM percentage calculated in each treatment was used for converting into t ha$^{-1}$. Dried samples were ground using a powerful grinder (FW-200, Beijing Zhong Xing Wei Ye Instrument Co., Ltd., Beijing, China) to pass a 1 mm sieve for further proximate and chemical analysis. The crude protein was analyzed using the Kjeldahl procedure [22]. The calculated data of crude protein percent were used for converting into t ha$^{-1}$.

### 2.4. Silage Preparation

Sub-samples of each fodder species were taken and used for silage preparation. For this aim, 3 kg of fresh mixture samples was taken from each plot and the samples without additives were manually pressed into the polythene bags. The dimensions of polythene bags were 0.6 m × 0.3 m. Fodder was filled into the plastic bag layer by layer, compacted every layer, by continuous trampling to remove air, and the plastic bags were sealed airtight and kept at room temperatures (~25 °C) to allow for anaerobic fermentation for 60 days. After the ensiling period, 500 g of mature silage samples was taken from the center of the ensiled mass of each plastic bag for further chemical analysis. The silage samples were air-dried at 105 °C for 24 h and the dried samples were ground to pass through a 1 mm screen for further silage quality, proximate and chemical analysis.

### 2.5. Determination of Nutritional Values of Green Fodder and Silage

The pH of silages was noted on the aqueous extract of silage by using a pH meter (Five Easy Plus FE28, Mettler Toledo Co., Ltd., Shanghai, China). The ground samples were ashed at 550 °C [23] for 2 h in a muffle furnace (Nabertherm, Lilienthal, Germany). The CP content was determined as N × 6.25 using the Kjeldahl procedure [22]. Ether extract (EE) was determined via a typical ether extraction method [24]. The neutral detergent fiber (NDF) and acid detergent fiber (ADF) contents were analyzed by using Van Soest procedures [25]. Chemical analysis of crude fiber (CF) was performed following the standard procedures of the Association of Official Analytical Chemists [22]. Nitrogen-free extract (NFE) was calculated by difference [100 − (CP% + EE% + CF% + total ash%)]. WSC was checked via the enthrone method, by using freeze dried samples, and extracted using water [26]. Minerals, including calcium (Ca), sodium (Na), potassium (K), phosphorus (P), and magnesium (Mg), were determined by atomic absorption spectrophotometry [22]. Ammonia-N concentration was determined according to the method of Broderick and Kang [27]. The content of organic acids, including (LA), acetic acid (AA), and butyric acid (BA), was measured by high-pressure liquid chromatography [28]. All other reagents were of analytical grade.

### 2.6. Statistical Analysis

Each test was carried out in triplicate. The data of fodder production and chemical analysis of different silages, scatterplot matrix, and Pearson correlation were analyzed using one-way-ANOVA on SPSS version 22.00 (IBM Co., Chicago, IL, USA), and the Duncan test ($p$-value 0.05) was used to compare the treatment means. Graphs were created on Excel 2010.

## 3. Results

### 3.1. Fresh Biomass Yield

The FBY of maize intercropped with soybean ranged from 31.2 to 38.6 t ha$^{-1}$ (Table 1). SM had a better FBY (46.2 t ha$^{-1}$) than other intercropped treatments. The harvesting dates also significantly influenced the biomass production, and the R3 stage (46.0 t ha$^{-1}$) yielded better than the R6 stage (36.8 t ha$^{-1}$). Significant differences in FBY were found between the years, and the average yield of the second year (43.2 t ha$^{-1}$) was higher than that of the previous year (39.1 t ha$^{-1}$).

### 3.2. Dry Matter Yield

The DMYs of maize and maize intercropped with soybean at different planting structures are shown in Table 1. SM had a better DMY (14.3 t ha$^{-1}$) than other intercropped treatments. Significant differences in DMY were found between the harvesting dates, and the R3 stage (14.2 t ha$^{-1}$) yielded better than the R6 stage (13.0 t ha$^{-1}$). The effect of years on DMY was also significant. DMY significantly differed in both years; the first-year DMY was higher (14.3 t ha$^{-1}$) than the second-year DMY (13.2 t ha$^{-1}$).

### 3.3. Crude Protein Yield

The maize mixed with soybean possessed better CPY (2.1–2.5 t ha$^{-1}$) than the SM (1.8 t ha$^{-1}$). Harvesting time also affected CPY, and the R3 stage (2.3 t ha$^{-1}$) had higher CPY than the R6 stage (1.9 t ha$^{-1}$, Table 1). Significant effects of years on CPY were also observed, and the average CPY in the second year was better than that in the first year.

### 3.4. Scatterplot Matrix Analysis of Maize and Maize-Soybean Intercrop Fodders

The scatterplot matrices of FBY, DMY, and CPY in maize and maize-soybean intercropped fodders are shown in Figure 3. FBY was highly positively correlated with DMY ($p < 0.05$), and negatively correlated with CPY ($p < 0.05$). DMY was negatively correlated with CPY ($p < 0.05$).

**Table 1.** Fresh biomass, dry matter and crude protein yield of maize and intercropped maize-soybean fodders.

| Treatment [(1)] | Fresh Biomass Yield | Dry Matter Yield | Crude Protein Yield |
|---|---|---|---|
| | (t ha$^{-1}$) | (t ha$^{-1}$) | (t ha$^{-1}$) |
| SM | 46.2a | 14.3a | 1.8c |
| 1M1S | 31.2d | 12.0c | 2.1b |
| 1M2S | 32.3d | 12.1c | 2.2b |
| 1M3S | 34.2c | 12.1c | 2.5a |
| 2M1S | 38.6b | 13.1b | 2.1b |
| SEM | 1.69 | 0.53 | 0.12 |
| LOS | ** | * | * |
| **Harvest time** | | | |
| R3 stage | 46.0a | 14.2a | 2.3a |
| R6 stage | 36.8b | 13.0b | 1.9b |
| SEM | 1.67 | 0.55 | 0.11 |
| LOS | ** | * | * |
| **Year** | | | |
| 2015 | 39.1b | 13.2b | 2.0b |
| 2016 | 43.2a | 14.3a | 2.4a |
| SEM | 1.66 | 0.53 | 0.14 |
| LOS | ** | * | * |

Note: [(1)] SM, sole crop maize; 1M1S, one-row maize to one-row soybean; 1M2S, one-row maize to two-row soybean; 1M3S, one-row maize to three-row soybean; 2M1S, two-row maize to one-row soybean; R3, reproductive milk stage; R6, reproductive maturity stage; SEM, standard error of means. LOS, level of significance * and **, significant differences at the 0.05 and 0.01 probability levels, respectively. Values followed by different letters are significantly different at the 0.05 probability level.

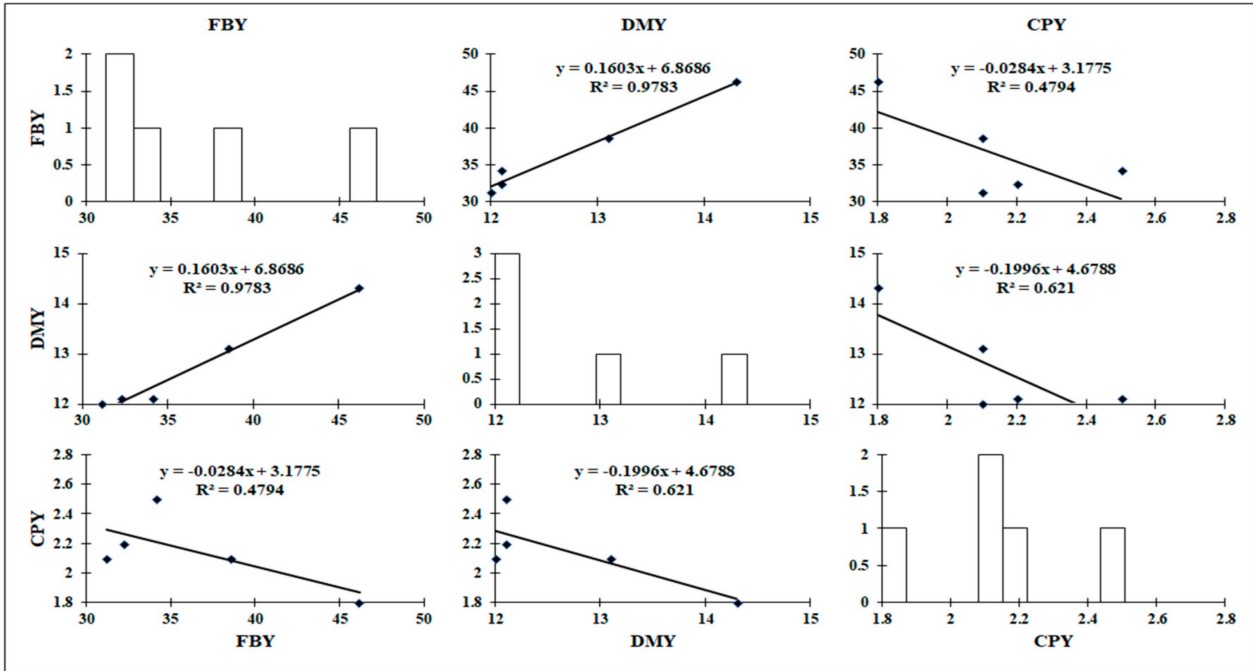

**Figure 3.** Scatterplot matrix including equation and regression values for pairwise correlation analyses between fresh biomass yield (FBY), dry matter yield (DMY) and crude protein yield (CPY) of maize and intercropped maize-soybean fodders. Diagonal boxes show histograms for each variable. The lower triangular matrix shows the relationship between a pair of variables.

### 3.5. Proximate Composition of Fodder

The proximate compositions of green fodders of maize and intercropped maize and soybean are shown in Table 2. The DM content of the intercropped fodder was significantly ($p < 0.01$) influenced by the harvesting time of the maturity stage compared with that of the maize fodder alone. The effect of the year was also significant, and the average DM content of the second year (41.2%) was higher than that of the first year (37.1%). The CP contents of maize intercropped with soybean at different planting patterns were ($p < 0.01$) higher than those of SM fodder. A significant difference was also found between harvesting dates, with the R3 stage having higher average CP% than the R6 stage. A significant effect of years on CP% was observed, and the average CP content in the second year was better than that in the previous year. The EE, ash, CF, and ADF contents did not significantly differ ($p > 0.05$) between the harvesting dates and the years. The highest NFE content was found in the harvesting time of the R3 stage. Harvesting time and year significantly affected the average NFE contents in fodder. The NDF and WSC contents significantly differed ($p < 0.05$) between the harvesting time and the years.

**Table 2.** Nutrient compositions of green fodders of maize and intercropped maize and soybean at two maturity stages (% DM) [1].

| Fodder [2] | Quality Parameters (%) | | | | | | | | |
|---|---|---|---|---|---|---|---|---|---|
| | DM | CP | EE | Ash | CF | NFE | NDF | ADF | WSC |
| SM | 32.1e | 8.7d | 2.0a | 6.8a | 28.8a | 58.2a | 43.1a | 24.3a | 10.3a |
| 1M1S | 36.1d | 11.1c | 2.1a | 6.7a | 28.6a | 48.7b | 40.2c | 24.1a | 8.9c |
| 1M2S | 38.2c | 11.1c | 2.0a | 6.8a | 28.6a | 48.7b | 40.3c | 24.1a | 9.0c |
| 1M3S | 40.0b | 12.4a | 2.2a | 6.7a | 28.6a | 48.8b | 40.9b | 24.2a | 9.3b |
| 2M1S | 42.1a | 12.1b | 2.1a | 6.7a | 28.6a | 48.6b | 40.4c | 24.1a | 9.1c |
| SEM | 0.42 | 0.21 | 0.12 | 0.02 | 0.62 | 0.41 | 0.42 | 0.33 | 0.41 |
| LOS | ** | ** | ns | ns | ns | ** | * | ns | * |
| **Harvest time** | | | | | | | | | |
| R3 stage | 36.2b | 12.1a | 2.1a | 6.7a | 28.6a | 58.6a | 43.2a | 24.2a | 10.2a |
| R6 stage | 40.3a | 8.6b | 2.0a | 6.8a | 28.7a | 48.3b | 40.1b | 24.1a | 9.1b |
| SEM | 0.44 | 0.23 | 0.13 | 0.02 | 0.61 | 0.39 | 0.41 | 0.31 | 0.41 |
| LOS | ** | ** | ns | ns | ns | ** | * | ns | * |
| **Year** | | | | | | | | | |
| 2015 | 37.1b | 11.1b | 2.1a | 6.8a | 28.7a | 48.7b | 41.2b | 24.0a | 9.4b |
| 2016 | 41.2a | 12.3a | 2.1a | 6.8a | 28.7a | 56.3a | 42.4a | 24.0a | 10.3a |
| SEM | 0.43 | 0.22 | 0.12 | 0.03 | 0.62 | 0.42 | 0.43 | 0.32 | 0.42 |
| LOS | ** | ** | ns | ns | ns | ** | * | ns | * |

Note: [1] DM, dry matter; CP, crude protein; EE, ether extract; CF, crude fiber; NFE, nitrogen free extract; NDF, neutral detergent fiber; ADF, acid detergent fiber; WSC, water soluble carbohydrate. [2] SM, sole crop maize; 1M1S, one-row maize to one-row soybean; 1M2S, one-row maize to two-row soybean; 1M3S, one-row maize to three-row soybean; 2M1S, two-row maize to one-row soybean; R3, reproductive milk stage; R6, reproductive maturity stage; SEM, standard error of means. LOS, level of significance. ns, not significant; * and **, significant differences at the 0.05 and 0.01 probability levels, respectively. Values followed by different letters are significantly different at the 0.05 probability level.

### 3.6. Fermentation Quality, Proximate, and Mineral Compositions of Silages

The results of fermentation quality of different silages are shown in Table 3. The intercropped silage significantly affected pH compared with the SM. Significant differences in pH were noted between the SM and intercrop silages ($p < 0.05$), with SM having a lower pH (3.8). Higher silage pH was found in the R3 stage than in the harvesting time of the R6 stage. The average pH in the second year (4.4) was higher than that in the first year (4.1). Higher organic acids (LA, AA, and BA) and ammonia-N ($p < 0.05$) were also found in the 1M3S silages than in other silages. Harvesting dates also affected the silage quality, and the R3 stage showed more favorable silage quality than the R6 stage. This two-year comparison revealed significantly different silage qualities and quantities.

**Table 3.** Fermentation quality of maize and intercropped maize-soybean silage on dry matter basis (% DM) [1] of two maturity stages.

| Silage [2] | pH | Organic Acids (mmol/L) | | | NH$_3$-N |
|---|---|---|---|---|---|
| | | LA | AA | BA | |
| SM | 3.8c | 9.0c | 9.2c | 2.1c | 8.0d |
| 1M1S | 4.1b | 11.1b | 10.2b | 2.1c | 10.1b |
| 1M2S | 4.2b | 11.2b | 10.5b | 2.9b | 10.1b |
| 1M3S | 4.4a | 12.1a | 13.1a | 3.1a | 10.6a |
| 2M1S | 4.1b | 11.1b | 10.3b | 2.1c | 9.1c |
| S.E.M | 0.02 | 0.41 | 0.23 | 0.14 | 0.24 |
| L.O.S | ** | * | * | * | * |
| Harvest time | | | | | |
| R3 stage | 4.3a | 11.1a | 10.1a | 3.0a | 10.1a |
| R6 stage | 3.7b | 9.0b | 9.1b | 2.2b | 8.1b |
| SEM | 0.02 | 0.41 | 0.22 | 0.14 | 0.22 |
| LOS | ** | * | * | * | * |
| Year | | | | | |
| 2015 | 4.1b | 10.1b | 10.1b | 2.9b | 8.6b |
| 2016 | 4.4a | 11.2a | 11.2a | 3.1a | 10.0a |
| SEM | 0.03 | 0.42 | 0.24 | 0.16 | 0.26 |
| LOS | ** | * | * | * | * |

Note: [1] LA, lactic acid; AA, acetic acid; BA, butyric acid; NH$_3$-N, ammonia-nitrogen. [2] SM, sole crop maize; 1M1S, one-row maize to one-row soybean; 1M2S, one-row maize to two-row soybean; 1M3S, one-row maize to three-row soybean; 2M1S, two-row maize to one-row soybean; R3, reproductive milk stage; R6, reproductive maturity stage; SEM, standard error of means; LOS, level of significance * and **, significant differences at the 0.05 and 0.01 probability levels, respectively. Values followed by different letters are significantly different at the 0.05 probability level.

The proximate and mineral composition of forage and silage are shown in Table 4. The CP content was higher ($p < 0.01$) in silage at the harvest time of the R3 stage, and the effect of the year was higher in the second year. The EE and CF contents between the forage and silage did not significantly differ. The effect of the year on the EE and CF contents also did not significantly differ. The NFE and NDF contents significantly increased ($p < 0.01$) in forage. The effect of the year on both contents was significant. Higher ADF and ash contents were found in silage. Moreover, a significant difference was not observed in mineral composition, especially in the percentage of P, Mg, and K contents of forage and silage. Additionally, P, Mg and K contents were not significantly different between the years. However, the Ca and Na contents of the intercropped silages were higher ($p < 0.05$) than those of the SM.

*3.7. Correlation Analysis of Nutritional and Fermentational Compositions of Forage and Silage of Maize and Intercropped Maize-Soybean*

Correlation analysis was performed to further understand the relationship between the nutritional and fermentational composition of forage and silage of maize and intercropped maize-soybean (Table 5). DM was significantly correlated ($p < 0.05$) with CP, while CP was negatively associated with CF. A significant negative correlation was observed in CP and CF. In the present study, the correlation showed that the EE content was significantly correlated ($p < 0.05$) with pH, LA, and NH$_3$-N. The ash content was strongly correlated ($p < 0.01$) with the Ca and Na contents. NFE was highly correlated ($p < 0.01$) with NDF. Moreover, NDF was significantly correlated ($p < 0.05$) with WSC, and ADF was strongly correlated ($p < 0.01$) with the K content.

**Table 4.** Proximate analysis and mineral compositions of forage and silage of maize and intercropped maize and soybean (% DM) at harvested time of two maturity stages.

| Treatments [(1)] | Parameters (%) [(2)] | | | | | | | | | | | |
|---|---|---|---|---|---|---|---|---|---|---|---|---|
| | CP | EE | Ash | CF | NFE | NDF | ADF | P | Ca | Mg | K | Na |
| SM | 8.8d | 2.0a | 6.5b | 28.8a | 58.4a | 43.2a | 22.3b | 0.31a | 0.28c | 0.21a | 2.31a | 0.15c |
| 1M1S | 11.2c | 2.1a | 6.6b | 28.7a | 48.6c | 40.4c | 20.1c | 0.31a | 0.30b | 0.21a | 2.30a | 0.16b |
| 1M2S | 11.2c | 2.2a | 6.6b | 28.6a | 48.6c | 40.6b | 22.2b | 0.32a | 0.31b | 0.22a | 2.31a | 0.16b |
| 1M3S | 12.3a | 2.2a | 6.8a | 28.7a | 48.8b | 40.7b | 24.1a | 0.32a | 0.35a | 0.22a | 2.32a | 0.18a |
| 2M1S | 11.9b | 2.1a | 6.6b | 28.6a | 48.6c | 40.3c | 22.3b | 0.31a | 0.31b | 0.21a | 2.31a | 0.16b |
| SEM | 0.22 | 0.14 | 0.09 | 0.61 | 0.42 | 0.43 | 0.34 | 0.07 | 0.08 | 0.02 | 0.06 | 0.02 |
| LOS | ** | ns | * | ns | ** | ** | * | ns | * | ns | ns | * |
| (% DM) | | | | | | | | | | | | |
| Forage | 8.9b | 2.0a | 6.6b | 28.7a | 58.6a | 43.2a | 22.2b | 0.31a | 0.30b | 0.21a | 2.30a | 0.16a |
| Silage | 12.3a | 2.1a | 6.8a | 28.8a | 48.7b | 40.1b | 24.1a | 0.31a | 0.35a | 0.20a | 2.31a | 0.18b |
| SEM | 0.21 | 0.12 | 0.11 | 0.62 | 0.41 | 0.42 | 0.32 | 0.05 | 0.06 | 0.02 | 0.06 | 0.02 |
| LOS | ** | ns | * | ns | ** | ** | * | ns | * | ns | ns | * |
| Year | | | | | | | | | | | | |
| 2015 | 8.8b | 2.1a | 6.6b | 28.7a | 48.7b | 40.1b | 22.3b | 0.31a | 0.33b | 0.21a | 2.31a | 0.15a |
| 2016 | 12.4a | 2.1a | 6.8a | 28.7a | 57.6a | 42.1a | 24.2a | 0.31a | 0.36a | 0.22a | 2.30a | 0.18b |
| SEM | 0.24 | 0.13 | 0.12 | 0.66 | 0.44 | 0.44 | 0.33 | 0.07 | 0.08 | 0.03 | 0.06 | 0.02 |
| LOS | ** | ns | * | ns | ** | ** | * | ns | * | ns | ns | * |

Note: [(1)] SM, sole crop maize; 1M1S, one-row maize to one-row soybean; 1M2S, one-row maize to two-row soybean; 1M3S, one-row maize to three-row soybean; 2M1S, two-row maize to one-row soybean; SEM, standard error of means; LOS, level of significance. [(2)] DM, dry matter; CP, crude protein; EE, ether extract; CF, crude fiber; NFE, nitrogen free extract; NDF, neutral detergent fiber; ADF, acid detergent fiber; P, phosphorus; Ca, calcium; Mg, magnesium; K, potassium; Na, sodium. ns, not significant; * and **, significant differences at the 0.05 and 0.01 probability levels, respectively. Values followed by different letters are significantly different at the 0.05 probability level.

**Table 5.** Pearson's correlation coefficient between nutritional and fermentational compositions of forage and silage of maize and intercropped maize-soybean on dry matter basis (% DM) of two maturity stages.

| | DM | CP | EE | Ash | CF | NFE | NDF | ADF | WSC | P | Ca | Mg | K | Na | pH | LA | AA | BA | NH$_3$−N |
|---|---|---|---|---|---|---|---|---|---|---|---|---|---|---|---|---|---|---|---|
| DM | 1 | | | | | | | | | | | | | | | | | | |
| CP | 0.91 * | 1 | | | | | | | | | | | | | | | | | |
| EE | 0.65 | 0.80 | 1 | | | | | | | | | | | | | | | | |
| Ash | 0.61 | 0.79 | 0.76 | 1 | | | | | | | | | | | | | | | |
| CF | −0.82 | −0.71 | −0.64 | −0.22 | 1 | | | | | | | | | | | | | | |
| NFE | −0.81 | −0.93 * | −0.79 | −0.60 | 0.81 | 1 | | | | | | | | | | | | | |
| NDF | −0.82 | −0.92 * | −0.72 | −0.53 | 0.82 | 0.99 ** | 1 | | | | | | | | | | | | |
| ADF | 0.35 | 0.25 | 0.38 | 0.60 | 0.00 | 0.05 | 0.12 | 1 | | | | | | | | | | | |
| WSC | −0.69 | −0.82 | −0.70 | −0.41 | 0.80 | 0.97 * | 0.98 * | 0.29 | 1 | | | | | | | | | | |
| P | 0.33 | 0.45 | 0.87 | 0.67 | −0.33 | −0.40 | −0.29 | 0.61 | −0.27 | 1 | | | | | | | | | |
| Ca | 0.71 | 0.84 | 0.82 | 0.98 ** | −0.35 | −0.64 | −0.58 | 0.65 | −0.45 | 0.72 | 1 | | | | | | | | |
| Mg | 0.33 | 0.45 | 0.87 | 0.67 | −0.33 | −0.40 | −0.29 | 0.61 | −0.27 | 1.00 ** | 0.72 | 1 | | | | | | | |
| K | 0.36 | 0.29 | 0.42 | 0.65 | 0.00 | 0.02 | 0.09 | 1.00 ** | 0.25 | 0.65 | 0.65 | 0.65 | 1 | | | | | | |
| Na | 0.61 | 0.79 | 0.76 | 1.00 ** | −0.22 | −0.60 | −0.53 | 0.60 | −0.41 | 0.67 | 0.98 ** | 0.67 | 0.65 | 1 | | | | | |
| pH | 0.73 | 0.90 * | 0.94 * | 0.93 * | −0.52 | −0.81 | −0.75 | 0.44 | −0.67 | 0.76 | 0.95 * | 0.76 | 0.49 | 0.93 * | 1 | | | | |
| LA | 0.81 | 0.97 ** | 0.89 * | 0.86 | −0.63 | −0.92 * | −0.88 * | 0.26 | −0.81 | 0.60 | 0.88 * | 0.60 | 0.31 | 0.86 | 0.97 * | 1 | | | |
| AA | 0.58 | 0.75 | 0.77 | 1.00 ** | −0.19 | −0.54 | −0.47 | 0.66 | −0.35 | 0.72 | 0.98 ** | 0.72 | 0.71 | 1.00 ** | 0.92 * | 0.83 | 1 | | |
| BA | 0.35 | 0.49 | 0.86 | 0.75 | −0.26 | −0.39 | −0.29 | 0.67 | −0.24 | 0.99 ** | 0.79 | 0.99 ** | 0.71 | 0.75 | 0.80 | 0.63 | 0.80 | 1 | |
| NH$_3$−N | 0.53 | 0.81 | 0.90 * | 0.80 | −0.47 | −0.84 | −0.78 | 0.11 | −0.78 | 0.68 | 0.78 | 0.68 | 0.17 | 0.80 | 0.92 * | 0.93 * | 0.77 | 0.69 | 1 |

Note: DM, dry matter; CP, crude protein; EE, ether extract; CF, crude fiber; NFE, nitrogen free extract; NDF, neutral detergent fiber; ADF, acid detergent fiber; WSC, water soluble carbohydrate; P, phosphorus; Ca, calcium; Mg, magnesium; K, potassium; Na, sodium; LA, Lactic Acid; AA, Acetic Acid; BA, Butyric Acid; NH$_3$-N, ammonia-N. The asterisks * and ** indicate the significant differences at the 0.05 and 0.01 probability levels, respectively.

## 4. Discussion

### 4.1. Comparison of Green Fodder, Dry Matter, and Crude Protein Yield of Intercropped Forage

The findings of this experiment are absolutely crucial in evaluating the yield and nutritional values of the intercropped forages of maize and soybean. The production of silage and its DM was found to be progressive. Fresh forage and DMYs were higher in the SM silages than in the four intercropped silages. Several researchers have reported variable results of intercropping systems. Geren et al. [29] indicated that mono-cropped maize produced higher biomass and DMY than maize intercropped with cowpea. Htet et al. [21] reported that, because of its tall and leafy structure, the maize in row intercropping had a markedly depressing effect on legume growth. Competition and unequal use of environ-

mental or underground resources, such as light and water, seem to account for problems experienced in intercropped communities. Htet et al. [30] indicated that these imbalances may have negative effects on crop yield by reducing leaves and leaf area index (LAI). Gou et al. [31] reported that intercropping with intercept photo synthetically active radiation increases production. Liu et al. [32] also revealed that the intercropping considered resource use as the biological basis for obtaining yield advantages. Yield advantages occur when the intercropped components compete only partly for the same plant growth resources, and interspecific competition is less than intra-specific competition [33]. Ideally, cultivars suitable for intercropping should enhance the complementary effects between species [21]. The previous research indicated that light, water, and nutrients are more completely absorbed and converted to crop biomass by intercropping. This finding is a result of differences in the competitive ability for growth factors between intercropped components [34]. Efficient utilization of available growth resources is fundamental in achieving sustainable systems of agricultural production. Eskandari et al. [34] pointed out that the density of mixed crops per unit area was the factor most closely correlated with the total yield of mixture and hence the factor most affected by competition. The FBY in the second-year was also higher than the first year, most probably due to the favorable summer conditions, including relatively high humidity, in 2016. Humidity increased progressively in the $C_4$ plant (maize) compared with the $C_3$ plant (soybean). The shading effect of soybean and groundnut by the maize plant (taller) may also have contributed to the reduction in the yield of intercropped legumes by reducing the photosynthetic rate of the lower growing plant [35]. Hunday and Hochman [36] suggested the above statement with their finding that water stress and shading contributed to the decrease in legume component yield under intercropping patterns. In their research on the agronomical properties of various crops, Jamshidi et al. [37] reported that DMY is crucial in evaluating any forages. In the present study, the average DMY of harvesting dates displayed a significant difference between R3 and R6 stages. This result may be due to the effect of competition in the dates of harvest. The average DMY in the second year was higher than the yields of the first year, probably because of the favorable climatic conditions in the second year, as discussed in biomass yield. Climate may also change on a yearly basis because of the environmental conditions.

Legume contributions to maize in mixtures are important sources of silage material in livestock husbandry [38]. Chang et al. [39] reported that silage pH and CP contents are the most dependable parameters of silage materials. The pH levels of silage mixtures were higher than those of maize silage, which had a favorable pH level. Although high pH levels in silage are not preferred in animal feeding, these pH levels were also within acceptable limits [40]. The published research showed that various types of legumes may increase the pH levels of silage, thereby causing a decrease in the quality of silage. Silage CP contents and yields were significantly different in the experiment of mixtures and monocropped maize. The mixtures possessed higher CP contents and yields. Altinok et al. [14] reported that sowing techniques are beneficial for silage CP content and yield.

### 4.2. Proximate and Nutrient Composition of Intercropped Forage and Silage

The most favorable DM range of the principal maize silage is between 28.5% and 32.2% [39]. The DM was linked to the fermentation conditions of the material [7]. Maize intercropped with cowpea (*Vigna unguiculata* L.) and climbing bean (*Phaseolus vulgaris* L.) produced a higher DM content than pure maize [29]. The results suggested that the contributions provided by legume components in the mixtures improved the CP yield of fresh fodder. The ratio of CP was increased in legume species when intercropping experiments were conducted in various crops.

The effect of soybean mixture in silage was estimated after fermentation because of its required time period. Legumes have larger pH and organic acid concentrations than grasses because legumes have greater buffering capacity [15,41]. Anil et al. [24] reported pH values of 3.7–4.6 and an average of 3.8 in silages of maize-soybean intercrops. The silage made from maize and soybean mixtures in the present research showed higher

LA and AA concentrations than the SM. Pakarinen et al. [42] also reported an increase in the LA contents when maize was ensiled in the mixture with other legumes. The results of intensive fermentation showed a lower DM content in the mixture than in the SM. Costa et al. [7] found 7.0–11.8% LA contents in the DM of maize silage. Anil et al. [24] added 30%, 40%, and 50% soybean in maize silages, and obtained LA contents of 4.8%, 5.0%, and 5.1% in the DM, respectively. The BA content in silage is also a sign of its high quality [43]. Mustafa et al. [44] reported silages of leguminous plants, which possessed BA contents lower than 0.5%. In the early maturity stage, low DM content was associated with butyric acid fermentation and large digestible nutrients. Increased DM content may cause limited fermentation and a decline in silage nutritional values [45]. In the present study, the LA content obviously decreased from the flowering stage to the dough stage, and this finding could be attributed to the decrease in moisture content upon maturity. Xie et al. [46] reported that the hydrolysis of starch in wheat grains by endogenous amylases could be the substrate for silage fermentation at the dough stage. Wilkinson and Davies [47] found that silage containing many organic acids and other volatile fatty acids is good for aerobic stability. Moreover, organic acids reduced the development of yeast and molds in silage. In the present study, 1M3S had a higher nutrient composition than the other intercropped silages.

The legumes are a rich and excellent source of protein contents, which gives silage a nutritional value for animal feeding practices. This study showed that the CP contents of intercropped silages 1M1S, 1M2S, 1M3S, and 2M1S were ($p < 0.05$) higher than those of the SM. The intercropping of maize with a variety of protein rich legume could improve the silage CP level by 3–5% and increase nitrogen digestibility, thus demonstrating a potential to decrease the requirement for purchased protein supplements [24]. The NDF contents of silages ranged from 29.7–40.3%. In the present study, the content of legume crops in the ensiled mass affected the NDF and ADF levels. A decreased fiber concentration in the DM of legumes was found in relation to grasses [7]. In addition, NDF level is related to the maturity stages of any crop with increased cell-wall components, especially cellulose, hemicelluloses, and lignin [48]. However, these effects have not been evaluated in other research, as no effect of intercropping was initiated on the NDF and ADF levels [7]. Saricicek et al. [49] found an increase in EE, ash, and CP contents in ensiled maize, with a decline in NDF contents after fermentation. However, the values obtained by these authors were contrary to what was obtained by Phiri et al. [50], who observed low CP and ash contents, whereas the decrease in NDF content was relatively in line with their results. Chaudhary et al. [51] stated that CP was negatively correlated with CF. CP was not only negatively correlated with NFE and NDF but also highly correlated with LA ($p < 0.01$). Ali et al. [52] reported that CP was negatively related with NDF. The increment of CP after fermentation may be attributed to the microbial synthesis of protein in the growth cycle of rumen and during the losses in carbohydrates. The decrease in NFE content could be attributed to the incomplete hydrolysis of WSC that advocated additional sugar for LA production during the fermentation process [24]. A previous study reported that the NDF content was decreased after the fermentation period [53], and the findings were comparable to those in the current experiment. The increase in ADF content after the ensiling period was in contrast to the report of Phiri et al. [50], who found that ensiling had no effect on ADF. The results showed that intercropping maize with soybean increased CP, and decreased NDF and ADF concentrations in silages. Therefore, SM silage is suggested to be at very high levels in obtaining high yields of fresh fodder and DM. Therefore, 1M3S was the best among the other intercropped silages according to the analysis of proximate in silage nutrient composition and nutritional values. WSC concentration increased in SM compared with that in intercrops. The highest WSC contents were found in SM, while sole cowpea produced the lowest DM digestibility [54]. Johnson et al. [54] also revealed an increase in WSC in stalks from tasseling to the milk stage and a turn-down thereafter. Other studies in maize have reported decreased ash content with the stage of maturity [55].

## 5. Conclusions

This study confirmed that the intercropping of maize with soybean at two maturity stages is an effective method to increase fresh fodder production and to enhance the nutrient quality of forage and silage, thus ensuring the supply of nutritionally rich forage and silage for livestock feeding. The general productivity of the system becomes more effective, and it could be advantageous for farmers in the area in terms of additive mixture.

In conclusion, maize and soybean intercropped as silage should be harvested at the milk (R3) stage to gain the highest CPY, and the lowest CF, NDF, and ADF contents. However, for high yield, SM silage is recommended. Finally, among all intercropped silages, the 1M3S was preferable according to proximate and nutrient composition compared to the other intercropped silages. Further investigation is required to determine how the management practices for intercropping maize and soybean could be optimized to produce silage of good quality and to evaluate the two or more maturity stages of forages for crop production and rich nutritional silage making.

**Author Contributions:** Conceptualization, M.N.S.H., J.-B.H., and B.-L.F.; Formal analysis, M.N.S.H. and J.-B.H.; Investigation, M.N.S.H., P.T.B., X.-W.G., C.-J.L., K.D., and L.-X.T.; Methodology, M.N.S.H. and J.-B.H.; Project administration, B.-L.F.; Writing—original draft, M.N.S.H.; Writing—review and editing, M.N.S.H., R.N.S., and K.L.A. All authors have read and agreed to the published version of the manuscript.

**Funding:** This research was funded by the Ministry of Science and Technology Innovation Team Project of Shaanxi Province, China (No. 2003AA102902), National Key Research and Development Program of China (nos. 2019YED1000700 and 2019YFD1000702), National Natural Science Foundation of China (31371529), National Science and Technology Supporting Plan (2014BAD07B03), Shaanxi Province Key Research and Development Project (2018TSCXL-NY-03-01), and Shaanxi Agricultural Collaborative Innovation and Extension Alliance Project (LMZD201803).

**Institutional Review Board Statement:** Not applicable.

**Informed Consent Statement:** Informed consent was obtained from all subjects involved in the study.

**Data Availability Statement:** The data presented in this study are available on request from the corresponding author.

**Conflicts of Interest:** The authors declare no conflict of interest.

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
