# Peer review of "Evaluation of Nutritive Values through Comparison of Forage Yield and Silage Quality of Mono-Cropped and Intercropped Maize-Soybean Harvested at Two Maturity Stages"

_agriculture, doi:10.3390/agriculture11050452_

Round 1

Reviewer 1 Report

Apart from the very low "number of replications" to assume results and thinking about reproducible methods, this paper, its discussion and conclusions, seem excessively generalist for this type of journal.

Author Response

Response to Reviewer 1 Comments

Comment: Apart from the very low "number of replications" to assume results and thinking about reproducible methods, this paper, its discussion and conclusions, seem excessively generalist for this type of journal.

First of all, on behalf of my co-authors, thank you very much for giving us an opportunity to revise our manuscript. We appreciate the positive and constructive comments. Based on reviewer suggestions and comments, we have thoroughly revised the paper. The list of responses to the comments is shown below. The main changes have been highlighted with red font in the revised manuscript. In the following replies, the red font represents the response, and the blue font means the text in the revised manuscript. We hope that it is now suitable for publication.

Dear reviewer, thanks for your comments. We appreciate the detailed and useful comments and suggestions from reviewer. It is really true as reviewer suggested that very low number of replications that is leads to lower precision in assuming the results. In order to have the best opportunity to detect differences, we should have at least 12 degree of freedom (df) for error. Higher df for error leads to lower Mean square (Error Mean square= Error Sum of Square/ df for error), consequently higher precision. Lower df for error means lower precision. We feel sorry for our carelessness in writing about experimental design in abstract section of our previous draft. According to reviewer suggestions and comments, we have been modified in revised manuscript (Please see sub title “2.2. Experimental Treatments and Design”). The sections of discussion and conclusions also have been modified in the revised manuscript. Please see detail of discussion in the revised manuscript. Also please see section of conclusions in the uploaded response file. Thank you again. 

Reviewer 2 Report

The corrections suggested for the English language have been made and are visible directly on the attached text

The main shortcoming of this paper is the bibliography which is not updated with the most recent evidence.

This makes it difficult to highlight the degree of innovation of the research presented and the scientific soudness.

Introduction: The introduction must be improved by integrating the background with the most relevant evidence from recent literature. Most of the bibliographic references are over ten years old.

Conclusion: Conclusion could be slightly narrowed down and made concrete with highlighting the future directions. Some statements of the conclusions should be moved to discussion

References

The bibliography must be supplemented with more recent studies.

Author Response

Response to Reviewer 2 Comments

Comment: The corrections suggested for the English language have been made and are visible directly on the attached text.

First of all, on behalf of my co-authors, we feel great thanks for your time in a professional reviews work on our article and thank you very much for giving us an opportunity to revise our manuscript. We appreciate the positive and constructive comments. Those comments are all valuable and very helpful for revising and improving our manuscript, as well as the important guiding significance to our researches. Based on your nice suggestions and comments, we have thoroughly revised the manuscript. The list of responses to the comments is shown below. The main changes have been highlighted with red font in the revised manuscript. In the following replies, the red font represents the response. We hope that it is now suitable for publication.

Dear reviewer, thank you for your comments. The language has been modified in the revised manuscript following the reviewer’s comments. Some incorrect statement has been deleted due to the deficiency of related proof. Thank you again.

Comment: The main shortcoming of this paper is the bibliography which is not updated with the most recent evidence.

Response: Dear reviewer, thank you for your comments and suggestions. We have reduced the number of old citations and references, and added update citations and references in the revised manuscript. All the detailed corrections could be seen in the reference section in which changes were marked by red. Thank you again.

Comment: This makes it difficult to highlight the degree of innovation of the research presented and the scientific soudness.

Response: Dear reviewer, thank you for your comments and suggestions. In our study, intercropping of maize with soybean for forage and silage is a feasible strategy to improve crude protein level. Compared with the ensilage of mono-crop maize, intercrops of maize and soybean for silage have higher proximate and mineral composition in feeding of animals. The best harvest time of intercropping is also an important issue in intercropped silage to get optimum nutritive compositions. As review suggested that animal nutrition focuses on studying the dietary needs of the animals. These dietary needs consist of nutrients, which are the components present in the feed that animals can digest and utilize. Hence, when feeding a diet, it is important to first its nutrient content. The results of the present study indicate that the general productivity of the system becomes more effective, and it could be advantageous for livestock farmers in the area in terms of additive mixture. The introduction and conclusions section have been modified in the revised manuscript. Thank you again.

Comment: Introduction: The introduction must be improved by integrating the background with the most relevant evidence from recent literature. Most of the bibliographic references are over ten years old.

Response: Dear reviewer, thank you for your insightful comments and suggestions. The introduction section has been modified in the revised manuscript and some incorrect statement has been deleted due to the deficiency of related proof. The update citations and references have been added in the revised manuscript. All the detailed corrections could be seen in the reference section in which changes were marked by red. Thank you again.

Comment: Conclusion: Conclusion could be slightly narrowed down and made concrete with highlighting the future directions. Some statements of the conclusions should be moved to discussion.

Response: Dear reviewer, thank you for your insightful comments and suggestions. The suggestion has been taken seriously and the conclusion section has been modified in the revised manuscript. Some statement of the result has been changed from the conclusion section to discussion section and added with relevant sentence or paragraph in the raised manuscript. Thank you again.

Comment: References: The bibliography must be supplemented with more recent studies.

Response: Dear reviewer, thank you for your insightful comments and suggestions. The new citations and references have been added in introduction, materials and methods, results and discussion. All the detailed corrections could be seen in the reference section in which changes were marked by red. Thank you again.

Reviewer 3 Report

Extend the Introduction chapter with knowledge from the point of view of animal requirements for feed composition. Some animals require a higher protein diet, others are less demanding. On the contrary, some animals are harmed by a protein-rich diet.

Figure 3 shows: The flow sheet of procedure used for making of fresh fodder and silage flours. Mark points to indicate when the chemical analyzes performed in Tables 2, 3 and 5 have been performed. I consider the lack of understanding of which part of the feed production process (presented in Figure 3) is to be attributed to the results given in Tables 2, 3 and 5 as a serious shortcoming.

Line 98 The article does not mention the methods for determining the parameters presented in Table 1 (N, P, K, OM). Complete it.

Line 140. Method for determining CP is missing. The CP parameter must be determined in each sample (not calculated).

Line 188. It is written: "FBY increased in SM, ..." I ask: Against what FBY increased? This sentence should be re-stylized.

Author Response

Response to Reviewer 3 Comments

Comment: Extend the Introduction chapter with knowledge from the point of view of animal requirements for feed composition. Some animals require a higher protein diet, others are less demanding. On the contrary, some animals are harmed by a protein-rich diet.

First of all, on behalf of my co-authors, we feel great thanks for your time in a professional reviews work on our article and thank you very much for giving us an opportunity to revise (major and minor) our manuscript. We appreciate the positive and constructive comments. Those comments are all valuable and very helpful for revising and improving our manuscript, as well as the important guiding significance to our researches. Based on your nice suggestions and comments, we have thoroughly revised the manuscript. The list of responses to the comments is shown below. The main changes have been highlighted with red font in the revised manuscript. In the following replies, the red font represents the response, and the blue font means the text in the revised manuscript. We hope that it is now suitable for publication.

Dear reviewer, thank you for your insightful suggestions. As reviewer suggested that animal performance or response is required in feeding of animals. Feeding has a direct impact on the growth rate, production capacity and health status of the animal as well as on the animal’s products quality. In addition to this, it also has effects on the environment. Therefore, knowledge on animal nutrition is key for a profitable and sustainable farming. Thanks for reviewer comment. It is the important deficiencies in this paper. We are very sorry to describe it clearly.

It is really true as reviewer suggested that some animals require a higher protein diet, others are less demanding. On the contrary, some animals are harmed by a protein-rich diet. Animal nutrition focuses on studying the dietary needs of the animals. These dietary needs consist of nutrients, which are the components present in the feed that animals can digest and utilize. Hence, when feeding a diet, it is important to first its nutrient content. The introduction chapter has been modified in the revised manuscript. Thank you again.

Comment: Figure 3 shows: The flow sheet of procedure used for making of fresh fodder and silage flours. Mark points to indicate when the chemical analyzes performed in Tables 2, 3 and 5 have been performed. I consider the lack of understanding of which part of the feed production process (presented in Figure 3) is to be attributed to the results given in Tables 2, 3 and 5 as a serious shortcoming.

Response: Dear reviewer, thank you for your kind suggestions. The figure 3 has been removed from the revised manuscript due to the deficiency of related proof. The details of field and laboratory works have been explained in the materials and methods. Please see the uploaded revised manuscript. Thank you again.

Comment: Line 98 The article does not mention the methods for determining the parameters presented in Table 1 (N, P, K, OM). Complete it.

Response: Dear reviewer, thank you for your insightful suggestions. The experimental site section has been modified in the revised manuscript following the reviewer’s comments (Please see in uploaded response file).Thank you again.

Comment: Line 140. Method for determining CP is missing. The CP parameter must be determined in each sample (not calculated).

Response: Dear reviewer, thank you for your comments. The method of CP parameter has been added under the sub section of “2.3. Plant Sampling and Fodder Production.” Please see the uploaded response file and thank you again.

Comment: Line 188. It is written: "FBY increased in SM, ..." I ask: Against what FBY increased? This sentence should be re-stylized.

Response: Dear reviewer, thank you for your comments. The sentence has been modified in the revised manuscript. Please see the uploaded response file and thank you again.

Reviewer 4 Report

The experience is quite well presented, the work is well documented. I recommend improvements in the methodology part to facilitate the understanding of the experience.
At Materials and Methods
I recommend the description of the soil type and its properties (NPK, humus), including the international equivalence of the name.
I recommend specifying the methods of data collection and the bibliography for the methods used in the experiment.
From my point of view, at subchapters should not be used abbreviations.

Author Response

Response to Reviewer 4 Comments

The experience is quite well presented, the work is well documented. I recommend improvements in the methodology part to facilitate the understanding of the experience.

First of all, on behalf of my co-authors, thank you very much for giving us an opportunity to revise our manuscript. We appreciate the positive and constructive comments. Those comments are all valuable and very helpful for revising and improving our manuscript, as well as the important guiding significance to our researches. Based on your nice suggestions and comments, we have thoroughly revised the manuscript. The list of responses to the comments is shown below.The main changes have been highlighted with red font in the revised manuscript. In the following replies, the red font represents the response, and the blue font means the text in the revised manuscript. We hope that it is now suitable for publication.

Dear reviewer, thank you for your insightful comments and suggestions. The methodology part has been modified in the revised manuscript. Thank you again.

Comment: At Materials and Methods I recommend the description of the soil type and its properties (NPK, humus), including the international equivalence of the name.

Response: Dear reviewer, thank you for your insightful suggestions. The experimental site section has been modified in the revised manuscript following the reviewer’s comments (Please see in uploaded response file).Thank you again.

Comment: I recommend specifying the methods of data collection and the bibliography for the methods used in the experiment.

Response: Dear reviewer, thank you for your insightful suggestions. The methods of data collection have been specified in the revised manuscript. We have reduced the number of old citations and references, and also added update citations and references in the revised manuscript. All the detailed corrections could be seen in the reference section in which changes were marked by red (please see the revised manuscript). Thank you again.

Comment: From my point of view, at subchapters should not be used abbreviations.

Response: Dear reviewer, thank you for your comments. The title of sub-section has been modified in the revised manuscript. Please see the uploaded response file and thank you again.

Round 2

Reviewer 1 Report

This paper has been reasonably improved

Author Response

Response to Reviewer 1 Comment (Round 2) Comments and Suggestions for Authors This paper has been reasonably improved Response: First of all, on behalf of my co-authors, we really appreciate you for your carefulness and conscientiousness. Dear reviewer, your suggestions are really valuable and helpful for revising and improving the scientific soundness and contribution of our research article. We feel great thanks to you for giving us an opportunity to revise our manuscript. Based on reviewer suggestions and comments, we have been thoroughly revised the whole manuscript including experimental design in methodology section, discussion and conclusions, when we first resubmitted the revised manuscript. Thank you again. We earnestly appreciate the Editor’s and Reviewers’ through work and hope that the revised manuscript will be met with approval. We hope that it is now suitable for publication. Once again, thank you very much for your careful review, comments and nice suggestions.
